# The 3-Year Effect of the Mediterranean Diet Intervention on Inflammatory Biomarkers Related to Cardiovascular Disease

**DOI:** 10.3390/biomedicines9080862

**Published:** 2021-07-22

**Authors:** Mireia Urpi-Sarda, Rosa Casas, Emilio Sacanella, Dolores Corella, Cristina Andrés-Lacueva, Rafael Llorach, Gloria Garrabou, Francesc Cardellach, Aleix Sala-Vila, Emilio Ros, Miguel Ruiz-Canela, Montserrat Fitó, Jordi Salas-Salvadó, Ramon Estruch

**Affiliations:** 1Department of Nutrition, Food Science and Gastronomy, Faculty of Pharmacy and Food Sciences, Campus Torribera, University of Barcelona, 08921 Sant Coloma de Gramanet, Spain; murpi@ub.edu (M.U.-S.); candres@ub.edu (C.A.-L.); rafallorach@ub.edu (R.L.); 2CIBER Fragilidad y Envejecimiento Saludable (CIBERFES), Instituto de Salud Carlos III, University of Barcelona, 08028 Barcelona, Spain; 3Department of Internal Medicine, Hospital Clinic de Barcelona, Institut d'Investigació Biomèdica August Pi i Sunyer (IDIBAPS), University of Barcelona, Villarroel 170, 08036 Barcelona, Spain; rcasas1@clinic.cat (R.C.); esacane@clinic.cat (E.S.); garrabou@clinic.cat (G.G.); FCARDELL@clinic.cat (F.C.); 4CIBER (Centro de Investigación Biomédica en Red) 06/03: Fisiopatología de la Obesidad y la Nutrición, Instituto de Salud Carlos III, 28029 Madrid, Spain; dolores.corella@uv.es (D.C.); mcanela@unav.es (M.R.-C.); mfito@imim.es (M.F.); jordi.salas@urv.cat (J.S.-S.); 5Department of Preventive Medicine, University of Valencia, 46100 Valencia, Spain; 6Centro de Investigación Biomédica en Red (CIBER) de Enfermedades Raras (CIBERER), 28029 Madrid, Spain; 7Lipid Clinic, Service of Endocrinology and Nutrition, Institut d’Investigacions Biomediques August Pi Sunyer (IDIBAPS), Hospital Clinic, 08036 Barcelona, Spain; ASALA@clinic.cat (A.S.-V.); eros@clinic.cat (E.R.); 8Department of Preventive Medicine and Public Health, School of Medicine, University of Navarra, 31009 Pamplona, Spain; 9Cardiovascular Risk and Nutrition (Regicor Study Group), Hospital del Mar Research Institute (IMIM), 08003 Barcelona, Spain; 10Universitat Rovira i Virgili, Departament de Bioquímica i Biotecnologia, Unitat de Nutrició Humana, 43201 Reus, Spain; 11Institut d'Investigació Sanitària Pere Virgili (IISPV), Hospital Universitari San Joan de Reus, 43204 Reus, Spain

**Keywords:** dietary pattern, nutrition, inflammation, gene expression, cardiovascular diseases

## Abstract

The intervention with the Mediterranean diet (MD) pattern has evidenced short-term anti-inflammatory effects, but little is known about its long-term anti-inflammatory properties at molecular level. This study aims to investigate the 3-year effect of MD interventions compared to low-fat diet (LFD) on changes on inflammatory biomarkers related to atherosclerosis in a free-living population with a high-risk of cardiovascular disease (CD). Participants (*n* = 285) in the PREDIMED trial were randomly assigned into three intervention groups: MD with extra-virgin olive oil (EVOO) or MD-Nuts, and a LFD. Fourteen plasma inflammatory biomarkers were determined by Luminex assays. An additional pilot study of gene expression (GE) was determined by RT-PCR in 35 participants. After 3 years, both MDs showed a significant reduction in the plasma levels of IL-1β, IL-6, IL-8, TNF-α, IFN-γ, hs-CRP, MCP-1, MIP-1β, RANTES, and ENA78 (*p* < 0.05; all). The decreased levels of IL-1β, IL-6, IL-8, and TNF-α after MD significantly differed from those in the LFD (*p* < 0.05). No significant changes were observed at the gene level after MD interventions, however, the GE of CXCR2 and CXCR3 tended to increase in the control LFD group (*p* = 0.09). This study supports the implementation of MD as a healthy long-term dietary pattern in the prevention of CD in populations at high cardiovascular risk.

## 1. Introduction

Atherosclerosis is the leading cause of death (31%) and the primary initiator of two major morbidities worldwide: Coronary and cerebrovascular diseases (http://www.who.int/cardiovascular_diseases/en/, accessed on 19 April 2021) [1]. Appearance and progression of atherosclerosis are related to the oxidation of lipids in low-density lipoprotein particles (LDL). Recent studies indicate that the cell surface receptor CD36 facilitates internalization of oxidized LDL (oxLDL) and intracellular conversion of oxLDL to cholesterol crystals which are able to activate the Nod-like receptor protein (NLRP) 3 inflammasome in vitro experiments through its bond to toll-like receptors (TLRs) [2,3,4,5,6]. This process leads to the recruitment of leukocytes and a higher expression of adhesion molecules such as E-selectin and VCAM-1 on the endothelial surface of the artery [7]. Leukocyte activation takes place by chemokines (monocyte chemoattractant protein-1 (MCP-1), regulated on activation normal T cells expressed and secreted (RANTES), interferon gamma-induced protein 10 (IP-10)) and chemokine receptors (CCR2, CCR5, CXCR2), which attract monocytes, neutrophils, dendritic cells (DCs) and T cells into the intima [5,8,9,10,11] and induce inflammasome activation (Nod-like receptor protein (NLRP) in macrophages [4,6]. This activation leads to the maturation of caspase-1 and the processing of its substrates, interleukin (IL)-1β and IL-18 [2,6,12], and finally IL-1β promotes the development of lipid plaque and also destabilizes it [6,13].

Animal and human in vitro and in vivo studies have reported that foods rich in saturated fatty acids (SFA) activate TLR2 and TLR4, whereas others rich in n-3 polyunsaturated fatty acids (PUFA) and/or polyphenols inhibit the expression of IL-1β in the inflammasome, which is modulated by TLRs activation [14,15,16], and inactivated by NF-κβ [15]. In addition, the activation of TLRs in adipose tissue may enhance the expression of the target genes related to proinflammatory cytokines and chemotatic chemokines [17].

Likewise, the Mediterranean diet (MD) and diets enriched with SFA, monounsaturated fatty acids (MUFA) or n-3 PUFA modulate biomarkers related to inflammation and its regulation of genes [18,19]. Thus, several short-term interventional studies with MD have shown an improvement in inflammatory biomarkers such as MCP-1, CRP, E and P-selectin, VCAM-1, ICAM-1, TNF-α, IL-6, IL-7, IL-8, IL-10, and IL-18 [18,20]. However, there is little evidence about how diet can directly modulate the regulation of genes related to inflammation and plasma levels of their derived proinflammatory markers at long-term.

The aim of the present study was to investigate the 3-year effect of an intervention with MD supplemented with extra-virgin olive oil (MD-EVOO) or mixed nuts (MD-Nuts) compared to a control low-fat diet (LFD) on 14 inflammatory circulating biomarkers related to the atherosclerotic process in a free-living population with a high risk of coronary heart disease. An additional pilot study was designed to assess the gene expression of 10 genes related with the atherosclerotic process at baseline and after 3 years of interventions in 35 participants (12 MD-EVOO, 12 MD-Nuts, 11 LFD).

## 2. Materials and Methods

An expanded Methods section is available in the Appendix A.

### 2.1. Subjects and Study Design

The PREDIMED study is a parallel-group, single-blind, multicenter, randomized, controlled 5-year clinical trial aimed to assess the effects of a MD-EVOO or MD-Nuts on the primary prevention of cardiovascular diseases compared with a control LFD (www.predimed.es, accessed on 26 April 2021) [21,22]. This sub-study used data collected from 285 consecutive participants from two centers (Hospital Clinic of Barcelona and University of Valencia) in whom we determined inflammatory marker concentrations (*n* = 285) related to different stages of atherosclerosis at baseline and after 3 years of intervention. The gene expression analysis was carried out in a pilot study with a subpopulation of 35 participants (12 in MD-VOO intervention group, 12 in MD-Nuts intervention group, and 11 in LFD group) at baseline and after 3 years of intervention. The Institutional Review Board of the centers approved the study protocol, and all the participants provided the signed informed consent.

### 2.2. Plasma Inflammatory Biomarkers

At baseline and after 3 years of follow-up, fasting blood was collected. Plasma was obtained after centrifugation of blood and stored at −80 °C until assay. Plasma concentrations of 14 inflammatory biomarkers related to different stages of the atherosclerotic process (Appendix A) were measured. IL-1β, IL-6, IL-8, IL-12p70, IL-18, TNF-α, MCP-1, regulated on activation normal T cell expressed and secreted (RANTES/CCL5), macrophage inflammatory protein (MIP-1β/CCL4), interferon gamma-induced protein 10 (IP-10/CXCL10), and interferon gamma (IFN-γ) were determined using the Bio-Plex Pro^TM^ cytokine, adhesion molecules, and chemokine assays (Bio-Rad Laboratories Inc., Hercules, CA, USA), and epithelial neutrophil-activating protein 78 (ENA78/CXCL5) and inducible T cell alpha chemoattractant (I-TAC/CXCL11) were determined using the VersaMAP^TM^ kits (R&D Systems, Abingdon, UK). Plasma samples were diluted 1:3 with the diluents provided for each assay by the manufacturer. Data from the reactions were acquired using the Luminex^®^ 100™System (Luminex, Austin, TX, USA) and the Bio-plex Manager 6.1 Software (Bio-Rad, Hercules, CA, USA). Concentrations were obtained by standard calibration curves. Results are shown in pg/mL. High-sensitive C-reactive protein (hs-CRP) was determined by standard enzyme-linked immunosorbent assays [21]. Intra- and inter-batch coefficients of variation were less than 8.9% and less than 13.8%, respectively.

### 2.3. Gene Expression Analysis

The expression of 10 genes related to inflammatory stages of atherosclerosis (Appendix A) and an endogenous control (GAPDH) were measured in PMBC. PMBC were submitted to RNA extraction using NucleoSpin^®^ RNA II (Macherey-Nagel, Düren, Germany). cDNA was synthesized using 1 g total RNA at a final volume of 50 μL, employing random hexamer priming. Samples were stored at −80 °C until use. Microfluidic cards (TaqMan^®^ Array Cards, Applied Biosystems, Foster City, CA, USA) were used to analyze the expression of the following genes in quadruplicate: Toll-like receptor 2 (TLR2), TLR4, TLR6, inflammasome (NLRP3), caspase-1 (CASP-1), IL-receptor 1 (IL1R1), chemokine (C-C motif) receptor 2 and 5 (CCR2 and CCR5), chemokine (C-X-C motif) receptor 2 and 3 (CXCR2 and CXCR3). Gene expression was analyzed with the ViiA 7 Real-Time PCR System (Applied Biosystems, Foster City, CA, USA). The cycling parameters used were those suggested by the manufacturer [23]. The comparative cycle threshold method was used to assess the relative gene expression. All the samples were normalized to the expression of the endogenous control GAPDH, and values were expressed as relative units. The inter-assay variation coefficient was less than 7.1%.

### 2.4. Other Clinical Measurement

At baseline and after 3 years, trained personnel performed anthropometric and blood pressure, body weight, body mass index (BMI), and lipid profile were measured in all the participants (Appendix A).

### 2.5. Statistical Analysis

The statistical analyses were performed with SPSS version 18.0 software (SPSS Inc., Chicago, IL, USA). Variables were presented as mean and standard deviation (SD) or standard error of the mean (SEM) or 95% confidence interval (CI), as appropriate. Categorical variables were expressed as percentages. Plasma inflammatory biomarkers had a skewed distribution (Kolmogorov and Levene tests), and were therefore transformed to their natural logarithm for analysis.

Repeated-measures ANOVA was used to compare 3-year changes in the inflammatory biomarkers and gene expression, testing the effects of interaction of two factors: Time as a within-participants factor with 2 levels (baseline and 3 y) and the groups of consumption (2 MD groups and control group) with an adjustment for age, gender, BMI, smoking status, physical activity, and drug use (aspirin and statins). The multiple contrasts were adjusted by a Bonferroni post hoc test. The Pearson’s correlation was used for univariate association among changes in variables. ANCOVA test adjusted for age, gender, BMI, smoking status, physical activity, and drug use (aspirin and statins) was applied to calculate differences between 3 years and baseline values for the cytokines and chemokines. Values with a *p*-value < 0.05 were considered significant.

## 3. Results

### 3.1. Characteristics of the Participants and Compliance to Intervention

The participants were 67 ± 5.5 years old and nearly one-third were men (Table 1). A high proportion of participants were overweight or obese (>90%), and 81.8% had hypertension, 70.5% had dyslipidemia, and 55.4% were diabetic. The three groups were well balanced regarding demographic characteristics, anthropometric parameters, and cardiovascular risk factors (Table 1).

Three-year changes of anthropometric measures, cardiovascular risk factors, physical activity, and food and nutrient intakes of the participants were shown in Appendix A, respectively. During the 3 years of diet intervention, body weight and BMI levels of the participants in the MD-EVOO diet group significantly decreased (*p* < 0.05; both), and diastolic blood pressure significantly reduced in the three intervention groups (*p* < 0.05; all) (Appendix A). Plasma glucose concentration fell in the MD-EVOO group (*p* = 0.027), while plasma total-cholesterol (*p* < 0.03; all) and triglycerides (*p* < 0.005; all) concentrations improved in the three groups. LDL-cholesterol levels improved in both MD groups (*p* < 0.040; both), while the LFD showed a significant reduction of HDL-cholesterol (*p* = 0.013).

Compliance and adherence to interventions were determined with the urinary 3-hydroxytyrosol (3-HT) as biomarker of EVOO consumption and plasma α-linolenic acid as biomarker of nuts consumption (Appendix A). Urinary 3-HT and plasma α-linolenic acid concentrations increased significantly in participants in the MD-EVOO and MD-Nuts consumption, respectively, compared with the control LFD and the other MD intervention (*p* < 0.05) indicating good adherence with supplemental foods. Additionally, Appendix A shows the consumption of food and nutrients among the study participants. As expected, EVOO and nut intakes significantly increased in the corresponding MD groups (*p* < 0.001; both), as well as the MUFA after MD-VOO consumption and PUFA, linoleic and α-linolenic acid after MD-Nuts consumption (*p* < 0.05).

### 3.2. Changes in Chemokines and Cytokines between Baseline and 3 Years of Intervention

Figure 1 and Figure 2 shows differences between 3 years and baseline and also, between-group differences in several inflammatory biomarkers. The baseline and 3-year values of chemokines and cytokines were shown in Appendix A. Participants following a long-term pattern with MDs showed significantly decreased plasma levels of MCP-1 (*p* < 0.001, both), MIP-1β (*p* < 0.031, both), ENA78 (*p* < 0.001, both), IL-1β (*p* < 0.025, both), IL-6 (*p* < 0.013, both), IL-8 (*p* < 0.025, both), IL-12p70 (*p* = 0.022, MD-nuts), TNF-α (*p* < 0.022, both), IFN-γ (*p* < 0.040, both), and hs-CRP (*p* < 0.018, both), while no significant changes were observed in these biomarkers in the LFD group (*p* > 0.05), except for hs-CRP that showed a trend to reduction (*p* = 0.063). The plasma levels of RANTES significantly decreased in the three intervention groups (*p* < 0.037, all). No significant changes were observed for plasma IP-10, I-TAC, and IL-18 levels (Figure 1 and Figure 2).

Compared with the LFD group, the two MD groups exhibited significantly lower plasma concentrations of IL-1β, IL-6, IL-8, TNF-α (*p* < 0.05, all). In addition, the MD-EVOO group had 4-fold lower levels of ENA78 (*p* = 0.063) and 6-fold lower levels of MCP-1 (*p* = 0.067) than the control group and the MD-Nuts group had 5-fold lower IFN-γ levels (*p* = 0.042) and 7-fold lower MCP-1 levels (*p* = 0.035) compared with the controls.

### 3.3. Pilot Study of Gene Expression Changes at Baseline and after 3 Years of Intervention

The mRNA expression of all the studied genes did not change significantly after MD and LFD interventions in this pilot study with 35 participants (Figure 3) (*p* > 0.05). However, in the adjusted model, the mRNA expression of CXCR2 and CXCR3 showed a slight trend of increase only in the control LFD (*p* = 0.09). In addition, in the non-adjusted model, the mRNA expression of TLR6, TLR2, CCR2, CCR5, CXCR3, and CXCR2 significantly increased after 3 years only in the control LFD (*p* < 0.05) (Appendix A) suggesting new niches of study with higher number of participants.

### 3.4. Correlation Analysis of Changes in Inflammatory Biomarkers

Correlations among the three intervention groups were calculated to determine the convergence of cytokines/chemokines. It was found that cytokines/chemokines exhibited different patterns of correlation among the study groups. In the MD groups, the four significant chemokines were significantly inter-correlated, but this correlation was not observed between ENA78 and both MCP-1 and MIP-1β in the LFD group (Table 2). Along the same line, when we correlated changes in chemokine and cytokine concentrations (Table 3), the control LFD group showed a different behavior compared with the MD groups. Thus, higher and significant correlations were observed between changes in RANTES and both changes in IFN-γ and IL-6 (*p* < 0.05; both) and between changes in ENA78 and changes in TNF-α, IL-1β, and IL-6 (*p* < 0.05; all) in both MD groups. Nevertheless, these correlations were not statistically significant in the LFD group (*p* > 0.05). Correlations between cytokines showed the same behavior in the three groups (Appendix A).

## 4. Discussion

After 3 years of intervention, both MD groups showed significant anti-inflammatory effects demonstrated by the diminution of plasma cytokine and chemokine concentrations (MCP-1, MIP-1β, RANTES, ENA78, IL-1β, IL-6, IL-8, TNF-α, IFN-γ, and hs-CRP) related to atherosclerosis, together with improvement in classical cardiovascular risk factors (body weight, blood pressure, plasma cholesterol concentration or lipid profile). On the other hand, the LFD control group showed significant changes of plasma MCP-1, IL-1β, IL-6, IL-8, TNF-α, and IFN-γ concentrations compared with both MD groups. Considering gene expression, no significant changes were observed between MD interventions and LFD control. However, the LFD control group showed a tendency to increase expression of CXCR2 and CXCR3 genes after 3 years, while this trend was not observed after MD interventions. This is the first work studying the gene expression related to inflammatory stages of atherosclerosis considering such long dietary treatments. Previous works demonstrated beneficial effects of MD on gene expression related to inflammation when administering diets during lower time periods such as several weeks [24,25].

One of the early stages of atherosclerosis is the deposition and oxidation of LDL particles in the arterial intima. This results in the recruitment of monocytes, which differentiate to macrophages upregulating the expression of TLRs, leading to activation of NF-κβ and expression of chemokines with pro-atherogenic effects [26] (Appendix A). Increased TLR activity combined with the uptake or intracellular formation of cholesterol crystals, the effects of neutrophil extracellular traps, flow dynamic properties such as an atheroprone flow or intraplaque hypoxia may lead to NLRP3 inflammasome activation [27,28]. Through the caspase-1 system, it generated active IL-1β and IL-18 from pro-IL-1β or pro-IL-18, which then consecutively leads to a stimulation of the acute phase reaction [28,29]. In our study, the three molecules (TNF-α, IFN-γ, and IL-1β) involved in the response of NF-κβ showed a similar behavior after 3 years of intervention with MD, compared with the LDF group, while no differences were found in either IL-12 or IL-18 after any of the interventions. Therefore, our study showed that only the IL-1β and not the IL-18 pathway may be affected by long-term interventions with MD.

Previous in vivo studies have observed low concentrations of plasma biomarkers (IL-1β, IFN-γ or TNF-α) after short-term interventions with MD patterns compared to a SFA diet [18,20]. Contrary to our results, Casas et al. [30] and Esposito et al. [31] showed a decrease in IL-18 concentrations after 1 and 2 years of intervention with MD, respectively.

The results of our study show that MD decreases MCP-1 concentrations after 3 years, whereas those subjects in the control group did not show changes in plasma MCP-1 concentrations. Similar findings have been observed after a short-term intervention with MD in humans [20]. Liu et al. [32] reported that the critical role of TLR2 in the progression of atherosclerosis seems to be associated with MCP-1 levels and macrophage recruitment to atherosclerotic lesions.

RANTES is among the most highly expressed chemokine at transcript and protein levels in platelets and is associated with high risk of coronary artery diseases [33]. Our results showed that plasma concentrations of RANTES decreased in both MDs and LFD groups, a fact that can be due to RANTES high affinity binding and signaling with different chemokine receptors including CCR1, CCR3, CCR4, and CCR5 [34]. Other intervention studies with diets rich in SFA and MUFA have shown similar results with decreases in RANTES expression with these two diets [35].

The two inflammatory receptors CXCR2 and CXCR3 have also been widely involved in the pathogenesis of atherosclerosis [36]. Some studies have pointed out that CXCR2 might be involved in the recruitment of neutrophils to the vessel wall during the initiation of atherosclerotic plaque, being the main receptor of IL-8, while CXCR3 and its ligands IP-10 and I-TAC may promote atherogenesis by regulating the recruitment and balance of effector T cells and Tregs [37]. Our results showed a tendency of increase in both CXCR2 and CXCR3 in the LFD group, but without statistically significant 3-year changes in plasma I-TAC, IP-10, and IL-8 concentrations although concentrations of IL-8 were significantly different between the LFD group and both MD intervention groups. Nevertheless, MD appears to have an immunomodulatory effect on the expression of pro-inflammatory genes along the IL-8 pathway due to the significantly low concentrations observed for this molecule compared with the LFD group. Previous in vitro studies have shown that microbial metabolites of nuts consumption (urolithins) could decrease IL-8 production in human aortic endothelial cells stimulated with TNF-α [38]. In the same line, phenolic compounds of EVOO have also been related to downregulation of pro-inflammatory factors as IL-8 and IL-6 in human primary osteoarthritis chondrocytes induced by LPS [39].

The review of Barbaresko et al. [18] and the MOLI-SANI study [40] showed evidences to support the hypothesis that Western-like or meat-based patterns are positively associated with low-grade inflammation, while vegetable- and fruit-based patterns such as MD, appeared to be inversely related to inflammatory biomarkers. In the present study, we observed a significant diminution of inflammatory markers (MCP-1, IL-1β, IL-6, IL-8, TNF-α, IFN-γ, RANTES, ENA78, and hs-CRP) after a long-term intervention with a MD. These data are in line with previously published results by our group at short-term (3 and 12 months) [30,41], in which the MD seemed to exert an anti-inflammatory and immunomodulating effect through a diminution of pro-inflammatory interleukins (IL-1, IL-6), CRP, TNF-α, MCP-1, and soluble endothelial adhesion molecules (sVCAM-1, sICAM-1, sE- and sP-Selectin).

Inflammatory cytokines have potent effects on chemokine production by endothelial cells (EC). Duchene et al. [42] observed elevated ENA78 and CXCL6 mRNA expression in response to IL-1β, while Rouselle et al. [43] reported that among all the cytokines, only IL-6 was positively and highly correlated with ENA78 expression (r = 0.76). These results are in accordance with our study since after the MD interventions ENA78 was significantly correlated with IL-6 and IL-1β (*p* < 0.05) and RANTES with IL-6 and IFN-γ (*p* < 0.05; both), while in the LFD group we do not observe significant correlations, suggesting that these diets have different behaviors in the inflammation process. Additionally, in studies with human ECs, the combined treatment of oxidized LDL or IL-1β leads to an increase in ENA78 [44] expression, while IL-1β and IL-18, but not oxLDL, stimulate chemokine expression as MIP-1β [45]. The correlation between IL-1β and MIP-1β was only observed after MD interventions (Table 3). On the other hand, we also found a positive correlation between ENA78 and different chemokines with both MDs (Table 2) which were not observed in the LFD group for MCP-1 and MIP-1β. Previous studies by Balamayooran et al. [44] supported the direct and indirect role of ENA78 in monocyte recruitment after finding MCP-1 concentrations and macrophage numbers to be significantly reduced in the lungs of ENA78^−/−^ mice. In this respect, neutrophils (secreted by ENA78) transmigrated from the intravascular space into inflammatory tissues throughout the basement membrane and then secrete MIP-1β [46]. All the present data suggest that MD counteracts inflammatory processes related to atherosclerosis in a global and coordinated way.

The strengths of this study are the design as a randomized controlled clinical trial, the high completion rates, the good compliance determined by objective biomarkers, and the high number of inflammatory biomarkers and genes studied, which are involved in different phases of atheroma plaque formation. The study has several limitations. First, the low number of participants in whom we could measure the gene expression. Second, participants in the LFD group, despite receiving a personalized intervention, barely changed their dietary habits. Third, fatty acids in plasma do not reflect the long-term intake as accurately as adipose tissue or red blood cells do [47].

## 5. Conclusions

MD seems to reduce inflammatory processes at different molecular levels and avoid changes in gene expression related to atherosclerosis, exerting an immunomodulatory and anti-inflammatory effect upon the different pathways involved in inflammation. Thus, MD achieves a long-term anti-inflammatory effect through the diminution of plasma TNF-α, IFN-γ, and IL-1β, which are molecules involved in the response of NF-κβ, and also reduces inflammation via IL-1β, rather than IL-18. In addition, MD may act at a molecular level through CXCR2 affecting IL-8 plasma levels rather than acting via CXCR3 due to no changes for I-TAC and IP-10. Further studies should confirm this hypothesis derived of our results.

The results of the present study support the implementation of MD as a healthy long-term dietary pattern in the prevention of cardiovascular diseases in populations at high cardiovascular risk.

## Figures and Tables

**Figure 1 biomedicines-09-00862-f001:**
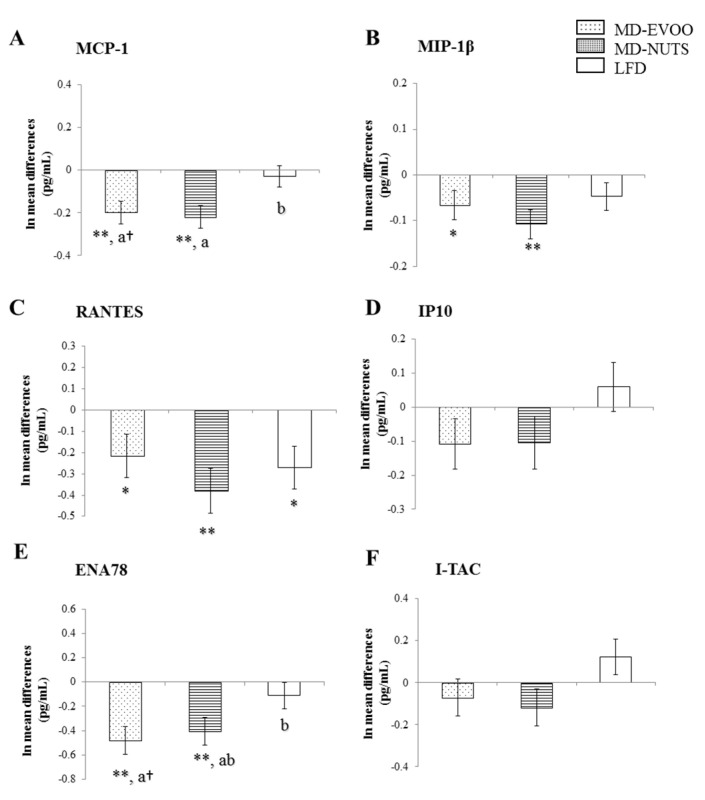
Mean change differences (SEM) between 3 years of intervention and baseline in chemokine plasma concentrations considering the groups of MD-EVOO (*n* = 93), MD-Nuts (*n* = 92), and control LFD (*n* = 100) in participants of the PREDIMED study. A to F, bar graphs show 3-year mean changes of MCP-1 (**A**), MIP-1β (**B**), RANTES (**C**), IP-10 (**D**), ENA78 (**E**), and I-TAC (**F**). Different from baseline, * *p* < 0.05; ** *p* ≤ 0.001; ^†^ *p* = 0.06–0.07 (Bonferroni post hoc test). ANCOVA was adjusted for age, gender, energy intake, BMI, smoking status, physical activity, and drug use. Mean change in a column with different superscripts letters are significantly different, *p* < 0.05 (Bonferroni post hoc test).

**Figure 2 biomedicines-09-00862-f002:**
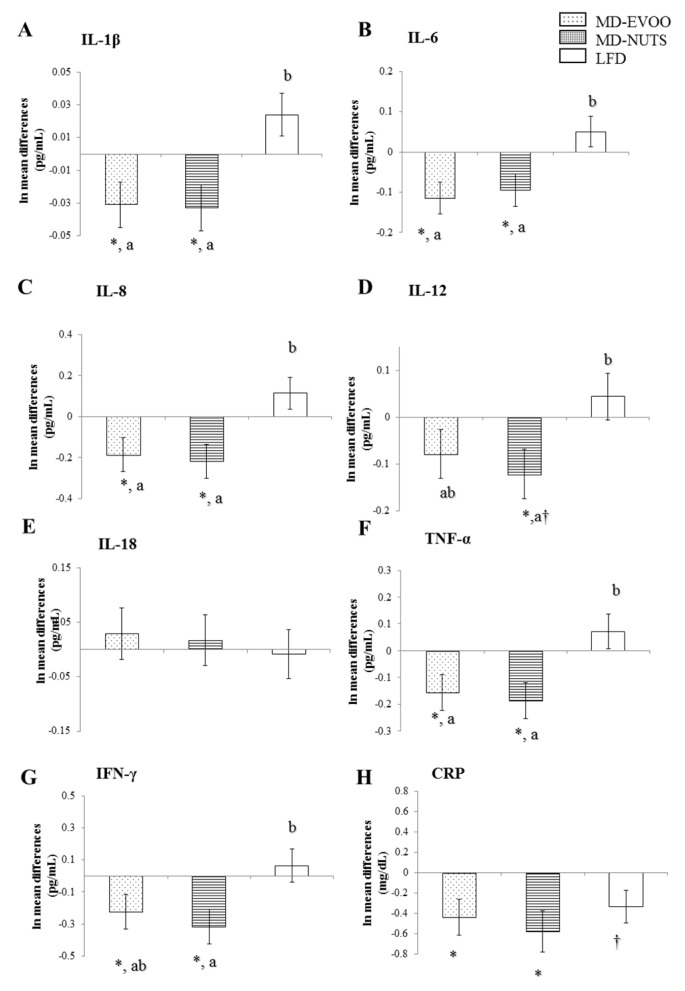
Mean change differences (SEM) between 3 years of intervention and baseline in cytokine and CRP plasma concentrations considering the groups of MD-EVOO (*n* = 93), MD-Nuts (*n* = 92), and control LFD (*n* = 100) in participants of the PREDIMED study. A to H, bar graphs show 3-year mean changes of IL-1β (**A**), IL-6 (**B**), IL-8 (**C**), IL-12p70 (**D**), IL-18 (**E**), TNF-α (**F**), IFN-γ (**G**), and hs-CRP (**H**). Different from baseline, * *p* < 0.05; ^†^
*p* = 0.05–0.06 (Bonferroni post hoc test). ANCOVA was adjusted for age, gender, energy intake, BMI, smoking status, physical activity, and drug use. Mean change in a column with different superscripts letters are significantly different, *p* < 0.05 (Bonferroni post hoc test).

**Figure 3 biomedicines-09-00862-f003:**
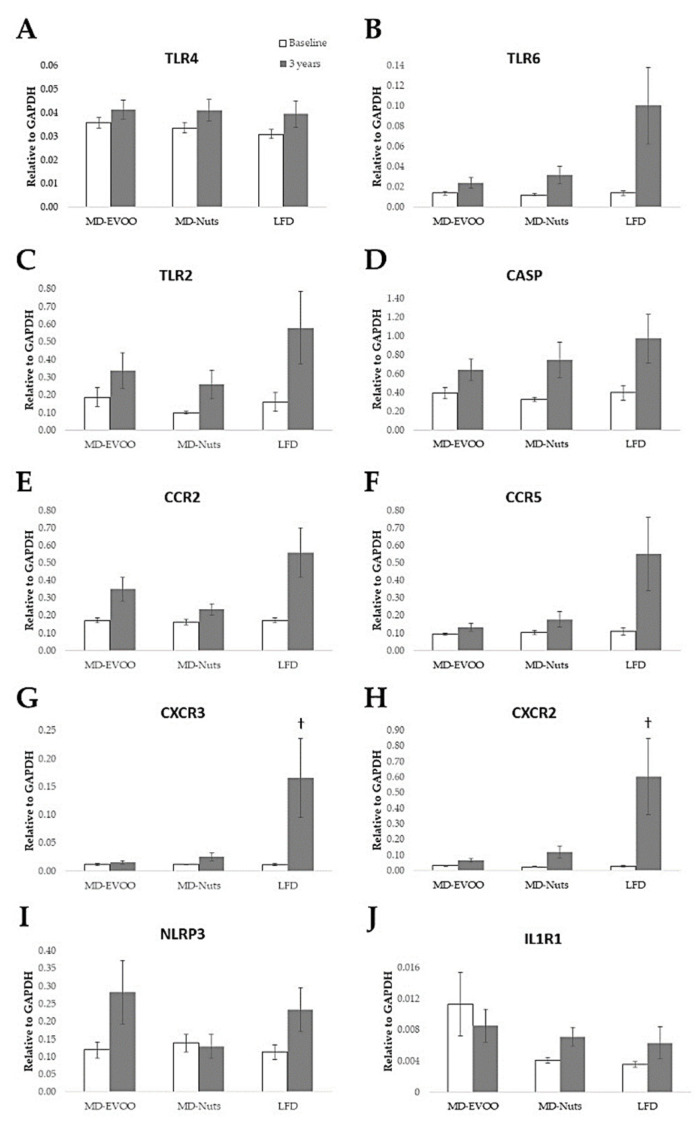
Expression of the genes TLR4 (**A**), TLR6 (**B**), TLR2 (**C**), CASP (**D**), CCR2 (**E**), CCR5 (**F**), NLRP3 (**G**), and IL1R1 (**J**) in participants of the PREDIMED study at baseline and after 3 years of intervention with MD-EVOO (*n* = 12), MD-Nuts (*n* = 12), and control LFD (*n* = 11). Different from baseline, ^†^ *p* = 0.08–0.09 (Bonferroni post hoc test). Repeated measures were adjusted for age, gender, energy intake, BMI, smoking status, physical activity, and drug use. Values are expressed as mean (SEM).

**Table 1 biomedicines-09-00862-t001:** Baseline characteristics of the 285 participants randomly assigned to the three intervention groups.

Characteristics	MD-EVOO (*n* = 93)	MD-Nuts (*n* = 92)	LFD (*n* = 100)	*p*-Value ^2^
Age, (years)	68 ± 6 ^1^	67 ± 5	67 ± 6	0.230
Men, *n* (%)	30 (32.3) ^1^	34 (37.0)	32 (32.0)	0.722
Family history of CHD, *n* (%)	34 (36.6)	19 (20.7)	23 (23.0)	0.066
Current smokers, *n* (%)	7 (7.5)	12 (13.0)	14 (14.0)	0.670
BMI (kg/m^2^)	30.9 ± 4.1	29.8 ± 4.0	30.5 ± 4.3	0.194
Waist circumference (cm)	103 ± 10.9	101 ± 10.2	102 ± 10.9	0.318
Systolic blood pressure (mmHg)	151 ± 21.1	154 ± 20.0	147 ± 18.2	0.111
Diastolic blood pressure (mmHg)	82.5 ± 9.9	86.1 ± 9.3	84.5 ± 10.1	0.127
Type 2 diabetes mellitus, *n* (%)	51 (54.8)	54 (58.7)	53 (53.0)	0.723
Hypertension, *n* (%)	76 (81.7)	76 (82.6)	81 (81.0)	0.959
Dyslipidemia, *n* (%)	62 (66.7)	64 (69.6)	75 (75.0)	0.434
Medication, *n* (%)				
ACE inhibitors	17 (18.3)	28 (30.4)	28 (28.0)	0.132
Diuretics	24 (25.8)	19 (20.7)	18 (18.0)	0.408
Other antihypertensive agents	45 (57.6)	36 (39.1)	47 (47.0)	0.395
Statins	34 (36.6)	24 (26.1)	48 (48.0)	0.007
Other-lipid-lowering agents	8 (8.6)	8 (8.7)	4 (4.0)	0.341
Insulin	6 (6.5)	7 (7.6)	10 (10.0)	0.652
Oral hypoglycemic drugs	31 (33.3)	25 (27.2)	37 (37.0)	0.344
Aspirin or antiplatelet drugs	10 (10.8)	19 (20.7)	16 (16)	0.181
NSAIDS	11 (11.8)	10 (10.9)	13 (13.0)	0.901

^1^ Values are mean ± SD or *n* (%) as appropriate. ^2^ From Pearson’s chi-square test for categorical variables and one-factor ANOVA for continuous variables. ACE: Angiotensin converted enzyme; CHD: Coronary heart disease; MD: Mediterranean diet; NSAIDS: Non-steroidal anti-inflammatory drug; EVOO: Extra-virgin olive oil.

**Table 2 biomedicines-09-00862-t002:** Pearson’s correlation of 3-year changes of chemokines in the three groups evaluated in the PREDIMED trial.

3-Year Changes	MIP-1β	RANTES	ENA78
	**MD-EVOO group**		
**MCP-1**	0.636 **	0.187 ^†^	0.441 **
**MIP-1β**		0.247 *	0.392 **
**RANTES**			0.444 **
	**MD-Nuts group**		
**MCP-1**	0.482 **	0.280 *	0.282 *
**MIP-1β**		0.513 **	0.322 *
**RANTES**			0.632 **
	**LFD group**		
**MCP-1**	0.387 **	0.308 *	0.124
**MIP-1β**		0.451 **	0.114
**RANTES**			0.494 **

**p* < 0.05; ***p* < 0.001; ^†^
*p* = 0.08.

**Table 3 biomedicines-09-00862-t003:** Pearson’s correlations of 3-year changes in the chemokines and cytokines studies in PREDIMED trial. Separated by the intervention group.

3-Year Changes	IFN-γ	TNF-α	IL-1β	IL-6	IL-8
	**MD-EVOO group**				
**MCP-1**	0.270 *	0.326 *	0.340 *	0.365 **	0.480 **
**MIP-1β**	0.175 ^†^	0.190	0.233 *	0.310 *	0.313 *
**RANTES**	0.270 *	0.122	0.074	0.295 *	0.190 ^†^
**ENA78**	0.528 **	0.393 **	0.316 *	0.369 *	0.471 **
	**MD-Nuts group**				
**MCP-1**	0.328 *	0.403 **	0.397 **	0.398 **	0.353 *
**MIP-1β**	0.217 *	0.116	0.233 *	0.301 *	0.230 *
**RANTES**	0.329 *	0.181	0.191 ^†^	0.336 *	0.301 *
**ENA78**	0.455 **	0.460 **	0.375 **	0.495 **	0.522 **
	**LFD group**				
**MCP-1**	0.323 *	0.241 *	0.218 *	0.343 *	0.403 **
**MIP-1β**	0.227 *	0.024	0.012	0.298 *	0.189 ^†^
**RANTES**	0.127	−0.044	0.009	0.020	0.193 ^†^
**ENA78**	0.280 *	0.194 †	0.089	0.127	0.319 *

* *p* < 0.05; ** *p* < 0.001; ^†^
*p* = 0.06–0.09.

## Data Availability

Not applicable.

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
