# Peer review of "The 3-Year Effect of the Mediterranean Diet Intervention on Inflammatory Biomarkers Related to Cardiovascular Disease"

_biomedicines, 2021, doi:10.3390/biomedicines9080862_

Round 1

Reviewer 1 Report

The authors investigated the effect of a diet rich in EVOO, nut or low fat component on CVD biomarkers in plasma in a 3-year study. The sample size, subject recruiting criteria and biomarker selection and determination is described in details and the research is within the scope of the journal. However, I don’t recommend its acceptance due to some issues in experiment design and data interpretation:

1) In this study, subjects with the diet of LFD were chosen as control but I think this should also be considered as a test group. Can the authors explain why a group without any dietary change was not used as the control?

2) Line 159. I can see ~1/3 of the subjects were male, not half. Do the authors believe there was an impact of gender on the result in this research?

3) Line 173-188 and supplementary table 1, 2 and 3.

It is hard to believe almost all the difference between groups are significant.

For example, in supplementary table 1, 77.2 +-13 was significantly different from 76.2 +-13.6 (p<0.05).

4) Figure 1 and 2.

In both figures, I can only read the data for those 3 groups after 3 years of intervention. The baseline is missing.

Author Response

RESPONSES TO REVIEWER #1

The authors investigated the effect of a diet rich in EVOO, nut or low fat component on CVD biomarkers in plasma in a 3-year study. The sample size, subject recruiting criteria and biomarker selection and determination is described in details and the research is within the scope of the journal. However, I don’t recommend its acceptance due to some issues in experiment design and data interpretation:

Comment 1. In this study, subjects with the diet of LFD were chosen as control but I think this should also be considered as a test group. Can the authors explain why a group without any dietary change was not used as the control?

Answer 1: We thank the review for this comment. PREDIMED study was designed in 2002-2003. Then, the usual care in that time it was a low-fat diet according to AHA guidelines. Thus, we started the trial with a usual care (control arm). However, when we published the results of the pilot study (Estruch et al., Ann Intern Med. 2006 Jul 4;145(1):1-11), the reviewers consider that the “usual care arm” (low-fat control diet) should receive an intervention with the same intensity of the other two arms. Thus, we modify our protocol and increased the intensity of the intervention in the low-fat control group.

Comment 2. Line 159. I can see ~1/3 of the subjects were male, not half. Do the authors believe there was an impact of gender on the result in this research?

Answer 2: We would thank the comment of the reviewer. We have changed half for one-third in the manuscript (page 4, line 167). Regarding the impact of the gender on the results, as gender is a key factor on cardiovascular risk (Du et al., Ann Med.2016;48(1-2):34-41; Cenko et al., J Am Coll Cardiol. 2019 Nov 12;74(19):2379-2389), we also adjusted the statistical models by gender. This information is stated on “Statistical Analysis” section (page 4, lines 155-164).

Comment 3. Line 173-188 and supplementary table 1, 2 and 3. It is hard to believe almost all the difference between groups are significant. For example, in supplementary table 1, 77.2 +-13 was significantly different from 76.2 +-13.6 (p<0.05).

Answer 3: We support the referee’s assertion that the changes are not very high, but they are statistically significant in the repeated-measures ANOVA analyses. For better understanding, we decided to express the differences between means with 95% confidence interval (CI). The nutritional and biochemical data observed in this sub-study go in the same direction of the changes observed in the overall population of the Predimed trial. Please see: Estruch et al., N Engl J Med. 2018;378(25): e34. doi: 10.1056/NEJMoa1800389; Urpi-Sarda et al., J Nutr. 2012; 142(6): 1019-25. doi: 10.3945/jn.111.148726.

Comment 4. Figure 1 and 2. In both figures, I can only read the data for those 3 groups after 3 years of intervention. The baseline is missing.

Answer 4: Figure 1 and 2 expressed the mean change differences between 3 years of intervention and baseline of cytokines and chemokines in plasma. We have rewritten the two figure captions to clarify it.

Additionally, in accordance to the reviewer, we have added a new supplementary Table (Table 4) with the baseline and 3-year values of cytokines and chemokines.

Reviewer 2 Report

The authors describe the 3-year effect of Mediterranean diet (MD) intervention compared to low-fat diet (LFD) as a sub-study of the PREDIMED study.

The authors found that intervention of MD reduce inflammatory levels and without changing gene expression.

Long-term effect of implementation of diet habit is striking issue especially from perspective of gene expression, so this paper is an interesting paper.

Author Response

RESPONSES TO REVIEWER #2

Comment 1. The authors describe the 3-year effect of Mediterranean diet (MD) intervention compared to low-fat diet (LFD) as a sub-study of the PREDIMED study. The authors found that intervention of MD reduce inflammatory levels and without changing gene expression. Long-term effect of implementation of diet habit is striking issue especially from perspective of gene expression, so this paper is an interesting paper.

Answer 1: We really appreciate the comment of this reviewer.

Reviewer 3 Report

The work by Urpi-Sarda and colleagues entitled “The 3-Year Effect of the Mediterranean Diet Intervention on Inflammatory Biomarkers Related to Cardiovascular Disease” describes the effect of Mediterranean dietary patterns (by the ingestion of virgin olive oil or nuts) on an extensive panel of inflammatory markers for a period of 3 years in a population at high cardiovascular risk. The groups are well balanced as seen by the anthropometric and biochemical parameters of individuals included in each group. Based on the data obtained authors conclude on the long-term beneficial effects of MD in a population at high cardiovascular risk. Conclusions are not novel though the extensive panel of inflammatory markers covered in this study prompts the suitability of the work for publication.

However, authors need to detail and clarify some points (please see below).

Materials and Methods Section

  1. In sub-section 2.1. (Subjects and Study design) states that the total number of individuals included in gene expression study was 35 though the total of individuals included in MD (12) and LFD (11) is different. Please clarify the correct number.

Results section

  1. How was the risk of cardiovascular diseases assessed? Was the biochemical parameter total cholesterol/HDL cholesterol the only one used? Or any others (carotida media intima thickness, coronary artery calcium score (CACS)?
  2. In light of data obtained and conclusions drawn, authors need to clarify on the cardiovascular risk which has not changed over the 3-year MD intervention as shown by the total cholesterol/HDL cholesterol values (Supplementary Table 1).
  3. As plasma cytokines improved for the three groups in the Predimed study regardless of dietary intervention (Supplemental Table 5) authors need to elaborate on the feasibility that improvement of inflammatory markers are the reflection of long-term hypocholesterolemic, hypotensive, and hypoglycemic drug therapies on plasma lipids (please see works by Bergheanu SC, Reijmers T, et al. (2008) Current Medical Research and Opinion 24: 2477–2487; Tarasov K, Ekroos K, et al. (2014) The Journal of Clinical Endocrinology and Metabolism 99: E45–E52; Hu C, Kong H, et al. (2011) Molecular BioSystems 7: 3271–3279; and Engelbrechtsen L, Lundgren J, et al., (2017) Obesity Sci & Pract 3(4), 425-433).
  4. Significance letter is used interchangeably as P or p throughout the text and tables (including Supplementary Material). Authors need to be coherent and consistent in their manuscript.

Author Response

RESPONSES TO REVIEWER #3

General Comment. The work by Urpi-Sarda and colleagues entitled “The 3-Year Effect of the Mediterranean Diet Intervention on Inflammatory Biomarkers Related to Cardiovascular Disease” describes the effect of Mediterranean dietary patterns (by the ingestion of virgin olive oil or nuts) on an extensive panel of inflammatory markers for a period of 3 years in a population at high cardiovascular risk. The groups are well balanced as seen by the anthropometric and biochemical parameters of individuals included in each group. Based on the data obtained authors conclude on the long-term beneficial effects of MD in a population at high cardiovascular risk. Conclusions are not novel though the extensive panel of inflammatory markers covered in this study prompts the suitability of the work for publication. However, authors need to detail and clarify some points (please see below).

Comment 1. Materials and Methods Section. In sub-section 2.1. (Subjects and Study design) states that the total number of individuals included in gene expression study was 35 though the total of individuals included in MD (12) and LFD (11) is different. Please clarify the correct number.

Answer 1: We thank the reviewer for the comment. We have changed the sentence to clarify the total of individuals included in each intervention group.

Now it reads “The gene expression analysis was carried out in a pilot study with a subpopulation of 35 participants (12 in MD-VOO intervention group, 12 in MD-Nuts intervention group and 11 in LFD group) at baseline and after 3 years of intervention.” (Page 3, lines 101-102). These changes have been highlighted in yellow in the manuscript file.

Comment 2. Results section. How was the risk of cardiovascular diseases assessed? Was the biochemical parameter total cholesterol/HDL cholesterol the only one used? Or any others (carotida media intima thickness, coronary artery calcium score (CACS)?

Answer 2: In the “Detailed and expanded methods” from the Supplementary Material, we specified the inclusion criteria of participants: At enrollement, participants did not have clinical cardiovascular disease, but they have a very high cardiovascular risk since they were type 2 diabetic or gathered at least three of the following major risk factors: smoking, hypertension (blood pressure ≥140/90 mm Hg or treatment with antihypertensive drugs), LDL-cholesterol concentrations ≥160 mg/dl (or treatment with hypolipidemic drugs), HDL-cholesterol ≤ 40 mg/d, body mass index (BMI) ≥25 kg/m2 , or a family history of early-onset coronary heart disease (CHD).

Therefore, we evaluated changes in the cardiovascular risk throughout repeated analyses of main biochemical parameters such as glucose and glycated haemoglobin, triglycerides, total-cholesterol, HDL-cholesterol and LDL-cholesterol and the ratio total cholesterol/HDL cholesterol. In addition, we also measured body weight, body mass index and adiposity parameters (waist perimeter), as well as, blood pressure (systolic and diastolic). These data are shown in Supplementary Table 1.

In this sub-study, unfortunately we did not measure carotid intima-media thickness or coronary artery calcium score. The effect of both supplemented Mediterranean Diets on internal carotid intima-media thickness and plaque height was assessed on other substudy of the Predimed trial and has been published by Sala-Vila et al., Changes in ultrasound-assessed carotid intima-media thickness and plaque with a Mediterranean diet: a substudy of the PREDIMED trial. Arterioscler Thromb Vasc Biol. 2014 Feb;34(2):439-45. doi: 10.1161/ATVBAHA.113.302327.

Comment 3. In light of data obtained and conclusions drawn, authors need to clarify on the cardiovascular risk which has not changed over the 3-year MD intervention as shown by the total cholesterol/HDL cholesterol values (Supplementary Table 1).

Answer 3: We would thank the reviewer for the comment. We agree that the total cholesterol/HDL cholesterol values did not change. However, other classical cardiovascular risk factors were improved in MD groups such as plasma glucose, triglycerides, LDL-cholesterol and total-cholesterol concentrations. In addition, blood pressure figures diminished in the two MD groups and BMI only in the MD supplemented with EVOO arm. Finally, we also should underline that HDL-cholesterol was significantly reduced in the control group.

Although we have more than 90 individuals in each group, it is possible that the absence of statistical significance observed in the analyses of some clinical and laboratory measurements may be due to a lack statistical power.

Comment 4. As plasma cytokines improved for the three groups in the Predimed study regardless of dietary intervention (Supplemental Table 5) authors need to elaborate on the feasibility that improvement of inflammatory markers are the reflection of long-term hypocholesterolemic, hypotensive, and hypoglycemic drug therapies on plasma lipids (please see works by Bergheanu SC, Reijmers T, et al. (2008) Current Medical Research and Opinion 24: 2477–2487; Tarasov K, Ekroos K, et al. (2014) The Journal of Clinical Endocrinology and Metabolism 99: E45–E52; Hu C, Kong H, et al. (2011) Molecular BioSystems 7: 3271–3279; and Engelbrechtsen L, Lundgren J, et al., (2017) Obesity Sci & Pract 3(4), 425-433).

Answer 4: We support the referee’s assertion that a long term hypocholesterolemic, hypotensive, and hypoglycemic drug therapies can improve inflammatory markers and plasma lipids, however, in the Predimed trial no significant changes were observed in medication used during follow-up by participants according to randomization group. See Table S6 – Supplemental Appendix in Estruch et al., N Engl J Med. 2018;378(25): e34. doi: 10.1056/NEJMoa1800389).

Nevertheless, we have adjusted statistical analysed by medication used (statins), since these drugs may affect plasma cytokines concentrations, in addition to lipid profile.

Comment 5. Significance letter is used interchangeably as P or p throughout the text and tables (including Supplementary Material). Authors need to be coherent and consistent in their manuscript.

Answer 5: We would thank the reviewer for the suggestion. We have changed all P or p by p, across all text, as the Biomedicines journal uses.